# On-the-Fly Ring-Polymer Molecular Dynamics Calculations of the Dissociative Photodetachment Process of the Oxalate Anion

**DOI:** 10.3390/molecules26237250

**Published:** 2021-11-29

**Authors:** Yukinobu Takahashi, Yu Hashimoto, Kohei Saito, Toshiyuki Takayanagi

**Affiliations:** Department of Chemistry, Saitama University, Shimo-Okubo 255, Sakura-ku, Saitama 338-8570, Japan; y.takahashi.895@ms.saitama-u.ac.jp (Y.T.); y.hashimoto.117@ms.saitama-u.ac.jp (Y.H.); k.saito.154@ms.saitama-u.ac.jp (K.S.)

**Keywords:** dissociative photodetachment, oxalate anion, reaction dynamics, ring-polymer molecular dynamics, path-integral molecular dynamics, nuclear quantum effect

## Abstract

The dissociative photodetachment dynamics of the oxalate anion, C_2_O_4_H^−^ + *h**ν* → CO_2_ + HOCO + e^−^, were theoretically studied using the on-the-fly path-integral and ring-polymer molecular dynamics methods, which can account for nuclear quantum effects at the density-functional theory level in order to compare with the recent experimental study using photoelectron–photofragment coincidence spectroscopy. To reduce computational time, the force acting on each bead of ring-polymer was approximately calculated from the first and second derivatives of the potential energy at the centroid position of the nuclei beads. We find that the calculated photoelectron spectrum qualitatively reproduces the experimental spectrum and that nuclear quantum effects are playing a role in determining spectral widths. The calculated coincidence spectrum is found to reasonably reproduce the experimental spectrum, indicating that a relatively large energy is partitioned into the relative kinetic energy between the CO_2_ and HOCO fragments. This is because photodetachment of the parent anion leads to Franck–Condon transition to the repulsive region of the neutral potential energy surface. We also find that the dissociation dynamics are slightly different between the two isomers of the C_2_O_4_H^−^ anion with closed- and open-form structures.

## 1. Introduction

Anion photodetachment spectroscopy is a powerful experimental technique for studying the nuclear dynamics on the potential energy surface of the corresponding neutral molecular system [1,2,3,4]. For example, if the stable structure of a monoanion is similar to the transition state structure of the neutral system, this technique can be employed to obtain detailed information on the properties of the transient transition state structure, including vibrational structures and energy distributions of the dissociated species. Thus, anion photodetachment spectroscopy can elucidate the nuclear dynamics on the potential energy surface region away from the stable potential energy well. Continetti et al. recently applied this technique to understanding the dissociative photodetachment dynamics of the oxalate anion (C_2_O_4_H^−^) and observed HOCO + CO_2_ products along with the relative translational energy distributions [5]. The oxalate anion is known to be stable with respect to electron loss, and this is in high contrast with the fact that the free oxalate dianion (C_2_O_4_^2^^−^) is unstable [6]. Thus, this is a simple example of the anion stabilization by solvation. The photodetachment experiment is one of the useful tools to understand the stabilization mechanism of such charged species. In the present work, we simulate the photodetachment dynamics of C_2_O_4_H^−^ theoretically to compare the computed results with their experimental results and to understand in greater detail the dissociation mechanisms and dynamics, which cannot be obtained from experiments alone.

The oxalate anion is a molecular system that contains CO_2_ molecular units with various charge states, which are important for understanding the activation of carbon dioxide [6]. As mentioned previously, the oxalate anion can be regarded as a complex of a proton (H^+^) and an oxalate dianion (C_2_O_4_^2^^−^) and has been previously studied experimentally and theoretically due to its unique structure [7,8,9,10,11,12,13,14,15]. The oxalate anion has two stable minima on the potential energy surface. One is a closed-form, in which the proton is asymmetrically bound between the two oxygen atoms, forming a planar structure with a five-membered cyclic ionic hydrogen bond, and this structure has an ideal geometry for studying proton-transfer dynamics between the two oxygen atoms [7,8,9,10,11,13]. The other is an open-form, in which the proton in the OH bond points out of the molecular framework with an O-C-C-O angle of 90°. Quantum chemical calculations show that the closed-form is about 0.4 eV more thermodynamically stable compared with the open-form and that the isomerization barrier via rotation of the OH group measured from the closed-form energy to the open-form energy of about 0.9 eV is high [12,14]. Thus, Continetti et al. concluded that the stable closed-form is preferentially produced under their molecular beam conditions [5]. They found that the detachment of C_2_O_4_H^−^ leads only to the two-body dissociation into HOCO + CO_2_ with a large kinetic energy release and that no other dissociation processes, including three-body dissociation, such as 2CO_2_ + H or CO_2_ + CO + OH, are observed. This result suggests that the dissociative potential energy surface along the C-C stretching coordinate is steeply repulsive for the neutral state. However, their experimental study could not identify the structure of the dissociated HOCO molecule, although HOCO can exist as two isomers, namely *trans*-HOCO and *cis*-HOCO. They speculated that *trans*-HOCO may be formed preferentially, based on their quantum chemistry calculation, in which geometry optimization starting from the minimum structure of the closed-form anion spontaneously produced CO_2_ and the *trans*-HOCO fragment. This also leads to an additional question on the correlation between the initial anion structure and the structure of the dissociated HOCO fragment. Therefore, it is interesting to study the dissociation dynamics also from the open-form C_2_O_4_H^−^ anion, although the open-form is thermodynamically unstable compared to the closed-form.

In this work, we report the results of quantum dynamics calculations using path-integral molecular dynamics (PIMD) and ring-polymer molecular dynamics (RPMD) [16,17,18,19,20], in which the anionic and neutral potential energy surfaces are directly obtained from on-the-fly quantum chemical calculations at the density-functional theory (DFT) level. Since both the PIMD and RPMD methods can describe nuclear quantum effects, these methods can reveal the importance of nuclear quantum effects in the photodetachment dynamics of C_2_O_4_H^−^. It is worth mentioning that the importance of nuclear quantum effects has been frequently pointed out in describing the photoabsorption and photoelectron spectra for various chemical systems [21,22,23,24]. In particular, it has been found that nuclear quantum effects are playing an essential role in hydrogen-bonded systems, where hydrogen atom motions behave quantum mechanically. Thus, it should be reasonable to apply the path-integral methods to understand the photodetachment dynamics of C_2_O_4_H^−^.

## 2. Computational Procedure

In the path-integral formalism, the classical isomorphic ring-polymer Hamiltonian for the chemical system consisting of *n* atoms is expressed as [16]
(1)$$H_{N}\left(P,Q\right)=\sum_{i=1}^{n}\sum_{s=1}^{N}\frac{{\left(P_{i}^{s}\right)}^{2}}{2m_{i}/N}+\sum_{i=1}^{n}\sum_{s=1}^{N}\frac{m_{i}N}{2\beta^{2}\hbar^{2}}{\left(Q_{i}^{s}-Q_{i}^{s-1}\right)}^{2}+\frac{1}{N}\sum_{s=1}^{N}V\left(Q^{s}\right),$$
where *N* is the number of imaginary time discretization steps (beads) and *β* is the reciprocal temperature with *β* = 1/*k_B_T*. *m_i_* is the atomic mass of the *i*-th nuclei. *Q_i_*^s^ and *P_i_*^s^ are position and momentum vectors of the *s*-th bead for the *i*-th atom, respectively. *V* is the potential energy surface. Classical sampling of the ring-polymer Hamiltonian preserves quantum Boltzmann statistics, where the corresponding equations of motion associated with the ring-polymer Hamiltonian are numerically solved using the first derivatives of the potential energy at each bead. In the sampling scheme, we use the standard PIMD approach, where the equations of motion are solved by controlling the system temperature with the massive Nóse–Hoover chain thermostat technique combined with the velocity Verlet algorithm. In general, the equations of motion are solved using the potential energy derivatives at each bead position.

To save computational time, the first derivatives, which are needed to solve the equations of motion, are approximately calculated with the second-order Taylor expansion at the centroid position of the beads as
(2)$${\left(\frac{\partial V}{\partial Q}\right)}_{Q^{s}}={\left(\frac{\partial V}{\partial Q}\right)}_{Q^{c}}+{\left(\frac{\partial^{2}V}{\partial Q^{2}}\right)}_{Q^{c}}\left(Q^{s}-Q^{c}\right)\;,$$
where *Q_s_* and *Q_c_* are the *s*-th bead position and centroid position, respectively. This approximation works well for *T* > 200 K, since the spread of quantum nuclei around the centroid is small. In this work, the PIMD calculations are performed at *T* = 300 K with 32 beads (Appendix A). The choice of these numerical parameters is coming from the benchmarking calculations performed with a different chemical system. Details are presented in the Supplementary Material. The total time step is set to 10^5^ with a time increment of Δ*t* = 0.3 fs. Then, the PIMD results are used to calculate the photoelectron spectrum. We use the following semiclassical approximation [25],
(3)$$I\left(E_{b}\right)\;\sim \;\int^{\;}\delta \left(E_{b}-V_{n}\left(Q\right)+V_{a}\left(Q\right)\right)\rho_{a}\left(Q\right)dQ$$
where *E_b_* is the electron binding energy and ***Q*** collectively denotes the structural coordinates. *V_n_*(***Q***) and *V_a_*(***Q***) are the potential energy surfaces for the neutral and anionic states, respectively. *ρ_a_*(***Q***) is the quantum nuclear density obtained from the PIMD simulation.

We subsequently perform RPMD simulations to obtain the real-time nuclear dynamics on the neutral potential energy surface with the same bead number (32 beads). The initial structures and momenta are chosen from the results of the PIMD simulations, and then, the equations of motion are solved similar to the PIMD simulation, but without the Nóse–Hoover thermostat. We choose 500 different initial conditions. At the end of each RPMD trajectory, the relative transitional energy between the HOCO and CO_2_ were calculated using the atomic centroid velocities. Similarly, we have calculated the H-O-C-O dihedral angle using the atomic centroid coordinates at the end of each RPMD trajectory to identify if the product HOCO molecule takes *trans* or *cis* configuration. The classical molecular dynamics calculations were also performed with the same scheme as the PIMD/RPMD calculations, except that the number of beads was set to one. In this case, vibrational quantization and quantum nuclear fluctuations are completely ignored. All the PIMD and RPMD calculations in this work are performed using the open-source PIMD code [26].

## 3. Results and Discussion

All the on-the-fly PIMD and RPMD calculations presented in this paper are performed at the B3LYP/aug-cc-pVDZ level theory including GD3 dispersion correction implemented in the Gaussian09 programs [27]. This DFT functional is chosen from the comparison to the CCSD(T)/aug-cc-pVDZ results [5]. More specifically, the energy difference between the most stable closed-form C_2_O_4_H^−^ anion and the neutral minimum with the open-form structure is calculated as 4.41 eV at the CCSD(T)/aug-cc-pVDZ level (without vibrational zero-point correction). The B3LYP/aug-cc-pVDZ level calculation provided this quantity as 4.45 eV. Notice that this energy is corresponding to the adiabatic electron affinity. In addition, the energy difference between the most stable closed-form anion and the open-form anion is calculated as 0.39 eV (without vibrational zero-point correction) at the CCSD(T)/aug-cc-pVDZ level, while this value was calculated as 0.38 eV at the B3LYP/aug-cc-pVDZ level. However, the B3LYP calculations yield slightly larger values for the energy levels of the asymptotic *cis*-HOCO + CO_2_ and *trans*-HOCO + CO_2_ fragments, which are 3.61 and 3.53 eV, respectively. The corresponding CCSD(T) values are reported to be 3.38 and 3.30 eV, respectively [6]. Thus, there is a possibility that the B3LYP potential energy surface gives smaller available energies for the dissociated fragments compared to the CCSD(T) potential energy surface.

Figure 1 displays the energy levels of the stationary points on the potential energy surfaces of C_2_O_4_H^−^ and C_2_O_4_H along with the energy levels of the neutral dissociated fragments. All the energy values are measured from the energy level of the most stable closed-form C_2_O_4_H^−^ anion. Figure 1 also shows the energy levels of the neutral state at the two anion minimum structures, which yield the vertical electron detachment energies. The neutral potential energy surface has three equilibrium structures. The first structure is the planar open-form with a C-C distance of 1.53 Å that can dissociate into CO_2_ + *cis*-HOCO via the transition state structure with a C-C distance of 1.58 Å. All the reaction pathways associated with the transition states found in this work were confirmed from the intrinsic reaction coordinate (IRC) calculations. The barrier height measured from the well of the planar open-form is calculated to be only 0.093 eV. The second minimum structure on the neutral potential energy surface corresponds to the hydrogen-bonded complex between CO_2_ and *trans*-HOCO with a long C-C distance (3.62 Å). The *C*_2*v*_ transition state is present, which is smoothly connected to the hydrogen-bonded minimum structure; there are two equivalent reaction paths leading to the hydrogen-bonded complex from the *C*_2*v*_ transition state. The third minimum structure on the neutral potential energy surface is another hydrogen-bonded complex between CO_2_ and *cis*-HOCO with a long C-C distance (3.60 Å). The energy levels of these two hydrogen-bonded complexes are just below the dissociation energy levels.

The anionic potential energy surface has two minima, corresponding to the open- and closed-forms. The C-C distance for the open-form (1.55 Å) is slightly shorter than that for the closed-form (1.59 Å), despite the presence of the five-membered cyclic structure. The results presented in Figure 1 qualitatively show the nuclear dynamics associated with the electron detachment process. When the excess electron of the most stable closed-form C_2_O_4_H^−^ is detached, direct dissociation may occur to form CO_2_ and *trans*-HOCO fragments. This is because the electron-detached Franck–Condon structure is at an energy much higher than the neutral *C*_2*v*_ transition state structure, and the electron detachment should be the transition to the almost repulsive region of the neutral potential energy surface. This scenario has already been discussed by Continetti et al. [5]. In contrast, if the excess electron is detached from metastable open-form C_2_O_4_H^−^, the dissociation can also occur, but via the transition state structure with the small barrier (Figure 1). This suggests the existence of an indirect dissociation mechanism, in which the nuclear trajectory may spend some time around the Franck–Condon region, which is followed by dissociation into CO_2_ + HOCO. Thus, there may be an important difference in the photodetachment nuclear dynamics between the closed- and open-form anions, although it may be difficult to preferentially produce metastable open-form C_2_O_4_H^−^ under experimental conditions. In fact, if we assume a Boltzmann equilibrium, the concentration ratio of the closed-form and open-form would roughly be 1:10^–6^ at *T* = 300 K.

Figure 2 shows the three-dimensional perspective plots of the nuclear distributions obtained from the PIMD calculations at *T* = 300 K for both the closed- and open-form C_2_O_4_H^−^ structures. Since there is a large barrier between these two forms (Figure 1), these two configurations can be independently sampled using different initial structural conditions. No isomerization between these two structures is observed within the present PIMD simulation time. For closed-form C_2_O_4_H^−^ (Figure 2a), the nuclear configuration is distributed around the planar structure. This observation indicates that the intra-molecular hydrogen bond is sufficiently strong to maintain the planar structure, and that electron detachment from closed-form C_2_O_4_H^−^ mainly produces the planar structure on the neutral potential energy surface. In addition, the proton is almost equally distributed around the two O atoms, indicating that proton transfer occurs between the two O atoms due to the small barrier, although the nuclear density in the proton-transfer transition state is small. For the open-form anion (Figure 2b), the O-C-C-O dihedral angle is broadly distributed around 90°, which is presumably due to the missing intramolecular hydrogen bond. This result suggests that electron detachment of the open-form anion yields neutral structures with a wide range of O-C-C-O dihedral angles.

In Figure 3a, we compare the photoelectron spectrum calculated with the semiclassical method based on the PIMD nuclear density with the experimentally measured spectrum taken from Ref. [5]. These spectra are plotted as a function of the electron binding energy with *E_b_* = *h**ν* − *eKE*, where *h**ν* and *eKE* are the excitation photon energy and kinetic energy of the photodetached electron, respectively. Here, *E_b_* corresponds to the energy level of the neutral state measured from the energy level of the most stable closed-form C_2_O_4_H^−^ (Figure 1). The experimental photoelectron spectrum is measured at a laser wavelength of 266 nm (*h**ν* = 4.66 eV) [5], and thus, the photoelectron spectrum in the higher electron binding energy region cannot be obtained. The spectrum calculated with the PIMD method shows a single broad peak about 0.5 eV wide at *E_b_* = 4.6 eV. Although the experimental peak position cannot be determined due to the limited photodetachment laser energy, our calculated spectrum qualitatively reproduces the experimental spectral features. Continetti et al. analyzed their measured spectrum with the bound-continuum Franck–Condon factor model calculated using two one-dimensional potential energy curves for the anionic and neutral states along the C-C stretch coordinate [5]. The simulated spectrum shows a spectral peak at *E_b_* ≈ 4.8 eV. If the one-dimensional analysis performed by Continetti et al. was correct, the peak position of our calculated spectrum should be shifted forward by about 0.2 eV. This indicates that the energy level of the neutral state calculated with the present DFT-B3LYP level is low; i.e., the electron affinity value is underestimated in the B3LYP calculations. Notice that good agreement in the vertical electron detachment energy is obtained between the CCSD(T) and B3LYP results, as mentioned previously. This indicates that more accurate calculations containing higher-order electron correlation must be performed to obtain more accurate electron affinity values.

In Figure 3, we also show the spectra calculated from the classical molecular dynamics, where the bead number was set to one. In this case, vibrational quantization is completely ignored. The calculated spectra show a narrower feature compared to the PIMD results. This indicates that nuclear quantum effects including quantum vibrational amplitudes are playing an essential role in determining the spectral shape. The present results are in line with the previous theoretical studies of the nuclear quantum effects on photoabsorption and photoelectron spectra [21,22,23,24].

Next, we present the simulated photoelectron–photofragment coincidence spectra. Figure 4 shows the simulated two-dimensional coincidence spectra plotted as a function of the kinetic energy release (*E_k_*) between CO_2_ and HOCO and the electron binding energy (*E_b_*). The simulated coincidence spectrum of Figure 4a can be compared with Figure 5 of Ref. [5], although the experimental spectrum is plotted as a function of *eKE* instead of *E_b_*. The simulated coincidence spectrum is obtained from 500 RPMD trajectories, and the kinetic energy release is calculated from the centroid velocities of the CO_2_ and HOCO dissociated fragments, as mentioned previously. There is a single broad Gaussian-like feature centered at *E_k_* = 0.9 eV and *E_b_* = 4.6 eV, which is qualitatively consistent with the experimental spectrum, although the experimental peak position for the kinetic energy release is observed at slightly higher energies of *E_k_* = 1.1 eV and *E_b_* = 4.7–4.8 eV. This difference in the peak position is originally coming from the fact that the B3LYP calculations give a lower energy level for the neutral state by about 0.2 eV at the Franck–Condon region. As mentioned previously, this energy difference affects the peak position (thus *E_b_*) of the photoelectron spectrum. In addition to this, it is emphasized that the energy level at the Franck–Codon region also affects the available energy for the HOCO and CO_2_ fragments; the vertical transition to a higher neutral state would lead to an increase in the available energy for the dissociation fragments. Another important reason for the difference in the peak position in the two-dimensional coincidence spectrum between theory and experiment is the inaccurate energy levels of the asymptotic HOCO and CO_2_ fragments. As mentioned previously, the B3LYP calculations give somewhat larger energy levels for both the *trans*-HOCO + CO_2_ and *cis*-HOCO + CO_2_ channels compared to the CCSD(T) energetics (see also Figure 1) [5]. This can lead to the decrease in *E_k_* in the simulated spectrum compared to the experimental result.

The simulated coincidence spectrum for the open-form anion obtained from 500 RPMD trajectories is also presented in Figure 4b, and it has similar features to that for the closed-form anion, although the peak is at a lower *E_b_* value.

To understand the detailed photodetachment dissociation mechanism of C_2_O_4_H^−^, we present the time dependence of the C-C distance and the relative velocity between CO_2_ and HOCO for all the RPMD trajectories examined in this work. The left and right panels of Figure 5 correspond to the results for the closed- and open-form anions, respectively. The results in Figure 5a,b show that the dissociation process for the closed-form anion has a strongly repulsive character because both the C-C distance and the relative velocity rapidly increase just after the electron detachment. In addition, the starting time of the flat region of the relative velocity shows that most of the dissociation processes are complete within *t* ≈ 60 fs. Only three RPMD trajectories out of 500 show an indirect, delayed dissociation mechanism. In contrast to the closed-form anion results, the open-form anion has different dynamics features. The relative velocity is nearly constant at an early stage up to *t* = 20–30 fs, and then, it increases rapidly after this region followed by a flat region at *t* > 75 fs (Figure 5d). This indicates that many trajectories spend time around the Franck–Condon region before fast dissociation occurs. These nuclear dynamics are consistent with the potential energy surface features of the neutral state. In addition, 10 RPMD trajectories out of 500 show the indirect mechanism, which is in contrast to the closed-form results. For the closed- and open-form anion, there is a step-like feature in the time dependence of the C-C distance at *t* ≈ 40 and 50 fs, respectively. These regions are close to the starting region of the flat velocity feature; thus, the step-like feature suggests that the trajectories reach the flat region of the potential energy surface at longer C-C distances. To understand the dissociation mechanisms in more detail, we present snapshots of typical RPMD trajectories in Figure 6. We did not observe the formation of hydrogen-bonded complexes, at least for the trajectories showing the direct dissociation mechanisms.

Finally, we identified the structure of the HOCO fragment by analyzing the H-O-C-O dihedral angular distributions. This analysis was performed at the end of each RPMD trajectory, where the relative velocity between CO_2_ and HOCO is almost constant (see also Figure 5). The black line in Figure 7 corresponds to the RPMD result calculated from the closed-form anion, while the red line corresponds to the result from the open-form anion. The former shows the angular distribution in the range of 150–210 degrees, and the latter shows the distribution from −45 to 45 degrees. From this result, it is found that dissociative photodetachment of the closed-form anion produces the *trans*-HOCO fragment and that photodetachment of the open-form anion gives the *cis*-HOCO product. In addition, these two angular distributions show a relatively narrow behavior. This result is consistent with fact that the dissociation dynamics is a fast direct process. In fact, from the results presented in Figure 5, dissociation of the closed-form completes within *t* < 60, fs while dissociation of the open-form completes within *t* < 75 fs. These results suggest that the available energy is not largely partitioned into the out-of-plane (internal rotational) motion of the OH bond in HOCO, and the *cis*−*trans* isomerization process does not occur in the dissociative photodetachment process. In other words, the *trans*-HOCO fragment can be formed preferentially from the closed-form anion, while *cis*-HOCO can be formed from the open-form anion. It is interesting that one of the two neutral HOCO isomers can be preferentially produced by choosing the initial anion structure through the dissociative photodetachment process.

## 4. Conclusions

We investigated the dissociative photodetachment of the oxalate anion, C_2_O_4_H^−^ + *h**ν* → CO_2_ + HOCO + e^−^, using the quantum PIMD and RPMD simulation methods, which can account for nuclear quantum effects. We used the on-the-fly technique, in which the potential energy derivatives are directly obtained from the DFT calculations at each time step. To reduce the computational time, the force acting on each bead was approximately calculated from the first and second derivatives of the potential energy at the centroid position of the nuclei beads. Since the anion potential energy surface has two isomers with closed- and open-form structures and the isomerization barrier between the two isomers is high, the PIMD and RPMD calculations for the two anion structures were independently performed. Only the results for the closed-form anion could be compared with the experimental results because the closed-form anion was more stable than the open-form anion. The photoelectron spectrum calculated using the PIMD method qualitatively agreed with the previous experimental spectrum [5], although the calculated spectral peak position was at a slightly lower energy compared with the experimental analysis. This is primarily due to the underestimation of the neutral energy level at the Franck–Condon region. The photoelectron spectra calculated using classical molecular dynamics method, which cannot describe quantized vibrational motions, show a narrower spectral feature compared to the PIMD results. This indicates that nuclear quantum effects are playing an essential role in determining the spectral shape.

We also compared the two-dimensional photoelectron–photofragment kinetic energy correlation spectrum calculated using the RPMD method with the previous experimental spectrum [5]. The calculated spectrum qualitatively reproduced the single Gaussian-like feature in the experimental spectrum, although the peak position is somewhat deviated from the experimental observation. Similar to the photoelectron spectrum case, this is due the inaccuracy of the DFT calculation, where the neutral energy level measured from the anion energy level is somewhat underestimated.

We discussed the difference in the photodetachment dynamics between the closed- and open-form anions, although no experimental data are available for the open-form anion. It is found that dissociative photodetachment of the closed-form anion preferentially leads to the *trans*-HOCO production and photodetachment of the open-form anion yields the *cis*-HOCO fragment. Detailed analysis of the RPMD trajectories shows that these dissociation processes are fast direct processes, which complete within *t* < 60–75 fs, and that the available energy is not largely partitioned into the out-of-plane motion of the OH bond in the HOCO fragment. We hope that the present computational study will stimulate future experimental studies of the open-form anion.

## Figures and Tables

**Figure 1 molecules-26-07250-f001:**
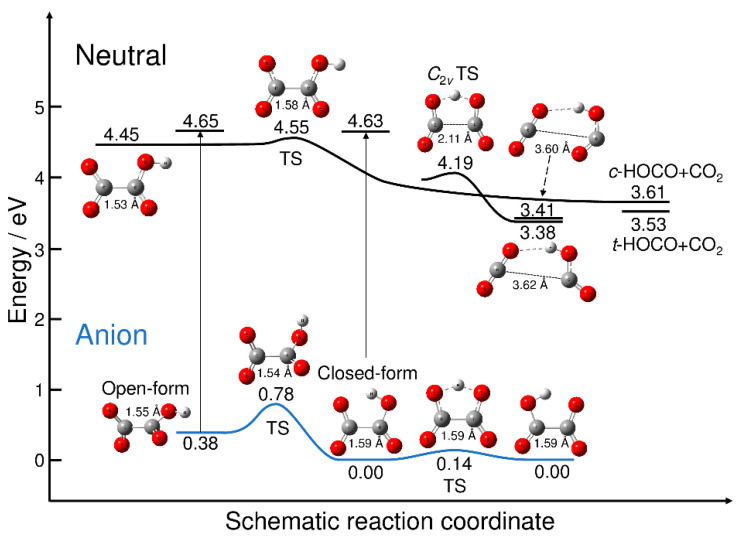
Schematic potential energy surfaces for C_2_O_4_H^−^ and C_2_O_4_H. Energy values were obtained at the B3LYP/aug-cc-pVDZ (with GD3 dispersion correction) level of theory. The energy was measured from the energy level of the most stable isomer of closed-form C_2_O_4_H^−^. The vertical thin lines with arrows indicate the vertical electron detachment process, and the corresponding energy values are the neutral energy levels at the anion minima. TS: transition state.

**Figure 2 molecules-26-07250-f002:**
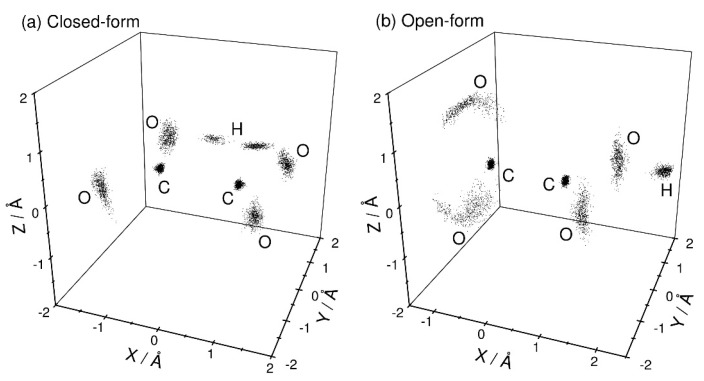
Three-dimensional perspective plot of the nuclear probability distribution functions of (**a**) closed-form and (**b**) open-form C_2_O_4_H^−^ calculated from the PIMD simulations at *T* = 300 K. The midpoint of the two carbon atoms is fixed to the coordinate origin, and the midpoint of the two oxygen atoms is always placed on the X-axis. In addition, the hydrogen atom is placed on the X-Y plane.

**Figure 3 molecules-26-07250-f003:**
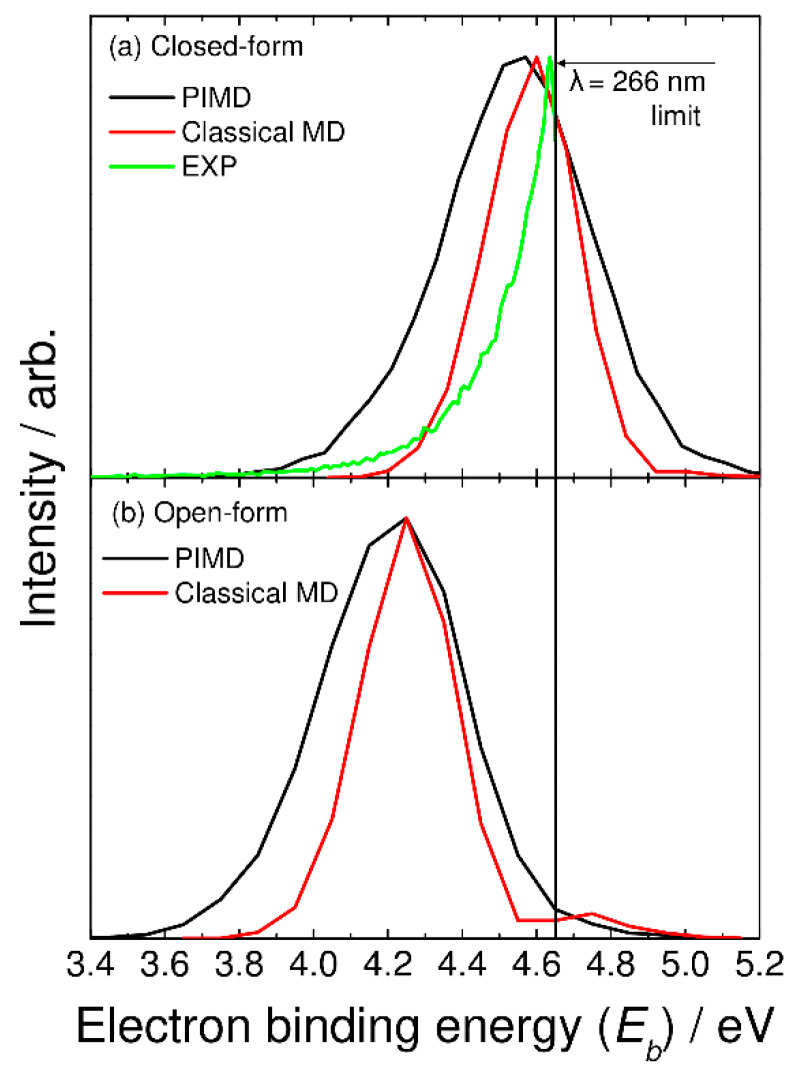
Photoelectron spectra for (**a**) closed-form and (**b**) open-form C_2_O_4_H^−^ calculated using the semiclassical method based on the PIMD nuclear density (black solid lines). In panel (**a**), the photoelectron spectrum calculated with the classical MD method is shown by the red solid line, and the experimental photoelectron spectrum measured at *λ* = 266 nm (taken from Ref. [5]) is also shown by the green solid line.

**Figure 4 molecules-26-07250-f004:**
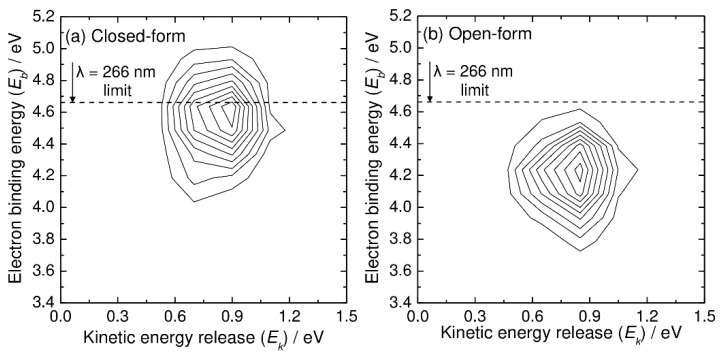
*N*(*E_k_*, *E_b_*) two-dimensional correlation spectra for C_2_O_4_H^−^ + *h**ν* → CO_2_ + HOCO + e^−^, where *E_k_* and *E_b_* are the relative kinetic energy between the CO_2_ and HOCO fragments and electron binding energy, respectively. Results from the RPMD calculations at *T* = 300 K for (**a**) closed-form and (**b**) open-form C_2_O_4_H^−^.

**Figure 5 molecules-26-07250-f005:**
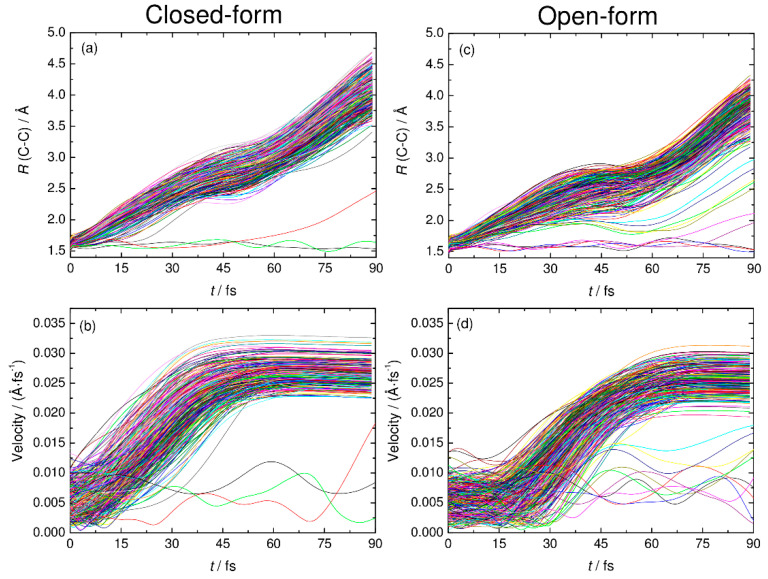
The C-C distance and relative velocity between CO_2_ and HOCO plotted as function time for all the RPMD trajectories examined in this work. Panels (**a**) and (**c**) show the results of the C-C distance for closed-form and open-form, respectively. Panels (**b**) and (**d**) show the results of relative velocity for closed-form and open-form, respectively.

**Figure 6 molecules-26-07250-f006:**
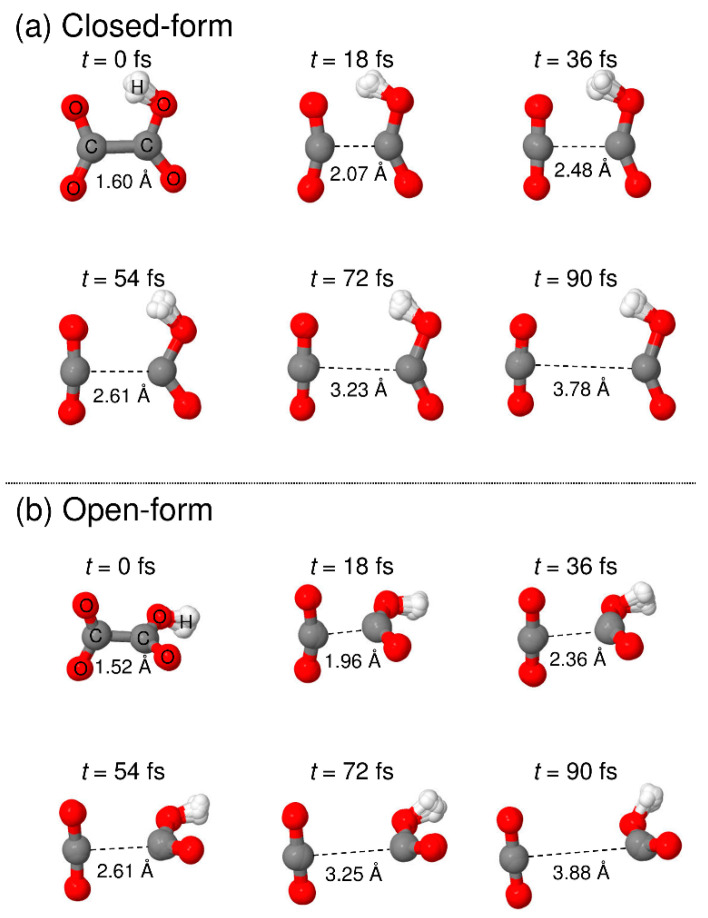
Snapshots of typical RPMD trajectories calculated starting from the (**a**) closed-form and (**b**) open-form C_2_O_4_H^−^ isomers.

**Figure 7 molecules-26-07250-f007:**
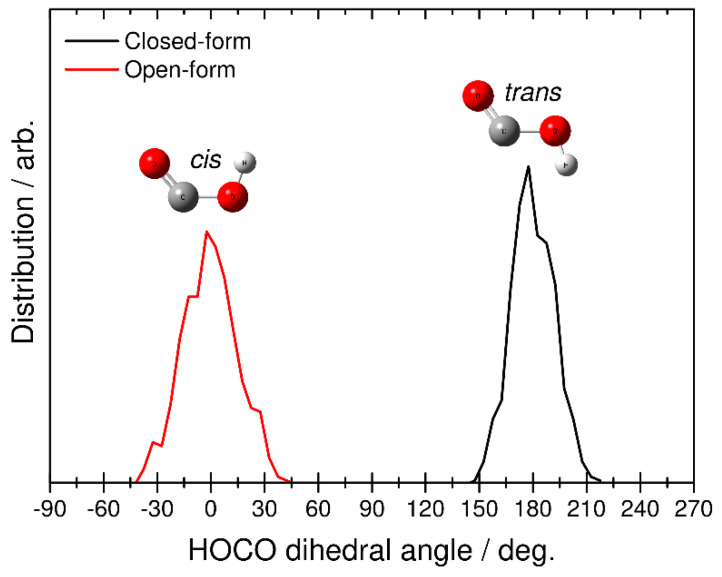
Distributions for the final HOCO dissociation product plotted as a function of the H-O-C-O dihedral angle obtained from the RPMD calculations. Black and red solid lines correspond to the RPMD results obtained from initial closed- and open-form C_2_O_4_H^−^, respectively.

## Data Availability

Not applicable.

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
