# Peer review of "On-the-Fly Ring-Polymer Molecular Dynamics Calculations of the Dissociative Photodetachment Process of the Oxalate Anion"

_molecules, 2021, doi:10.3390/molecules26237250_

Round 1

Reviewer 1 Report

This manuscript present a dynamical study on the photodetachment process of the oxalate anion, and its dissociation. The method used to study the dynamics is a combined PIMD/RPMD method, calculating on-the flight the potential, and its first and second derivatives, at the position of the centroid, using a DFT functional (B3LYP).The results are compared to experimental results for the photoelectron spectrum, showing a resonable agreement. Finall, it is demonstrated that the cis/trans isomers of the  HOCO products are directly obtained from the open/closed forms of the parent anion. The methods used are up to date, and the results are interesting. For all this, I consider this manuscript as publishable, after some points are addressed:

1) It is said that the B3LYP functional is chosen from a comparison with CCSD(T) results, from Ref. 5. A more detailed description of this comparison should be addressed and described.

2) In fig. 1 the shematic potential energy surface are ploted. What do they correspond to? Are they the minimum energy path or intrinsec reaction coordinate?. Also, the horizontal lines with a number, seems to indicate that this is the energy of the neutral species at the equilibirum energies of the anion, but this need to be specified.

3) How many beads are used for PIMD and RPMD calculations? What convergence test have been made.

4) Assuming a Boltzmann equilibrium, what is the relative partition functionbs of open and closed forms?

5) In Fig. 3 "Classical MD" results are shown, however they are not mentioned in the text, nor described. They must.

6) The experimental results shown in Fig.3 seems to indicate that the B3LYP

energy diffeence between neutral and anionic systems is underestimated. Some comments on this should be added (what is related to question 1)

Reviewer 2 Report

This paper investigates the dissociative photodetachment process of the oxalate anion via molecular dynamics method.  The difference in the photodetachment dynamics between the closed- and open-form anions is discussed.  Overall, the paper provides valuable findings on the dissociative photodetachment process of the oxalate anion at a molecular level, but there are still some comments to be addressed before I could recommend the paper for acceptance, as provided subsequently.

  1. In Abstract, please add the impact of the findings in this work.
  2. “Introduction”:
  • Paragraph 1: Please describe the significance of investigating the dissociative photodetachment of oxalate anion in real applications.
  • Paragraph 2: As the closed-form oxalate anion is more stable, please describe the significance of investigating the open-form oxalate anion.
  • Paragraph 3: Please review previous studies using PIMD and RPMD in describing the nuclear quantum effects in the photodetachment dynamics. As the two forms of oxalate anion have been investigated in previous works, please emphasize the aim of this work.
  1. “Computational procedure”:
  • Paragraph 3: the force acting on each bead was approximately calculated from the “first derivative” or “first and second derivatives” as described in Abstract? Is this approximation reasonable?
  • Please enrich the description of the photodetachment process simulation and the measurement of the two forms of oxalate anion.
  1. “Results and discussion”:
  • Figure 3: Please describe the reason for the underestimation of the energy level in this work compared with the experiment.
  • Figure 4: Please describe the comparison between Fig. 4(a) and the figure in Ref. 5 as stated in the manuscript. Please explain why the energy levels are lower and channels are smaller compared with the experiment.
  • Figure 7: Please enrich the description of the relationship between Fig. 7 and the dissociation time within 60-75 ns.
  1. “Conclusions”:
  • Please enrich the description of the detailed mechanism in Section 3.

Round 2

Reviewer 2 Report

The authors have addressed all the comments.  I would recommend the paper for publication.